# Effect of Sn Content on the Microstructure, Mechanical Properties and Corrosion Behavior of Biodegradable Mg–x (1, 3 and 5 wt.%) Sn–1Zn–0.5Ca Alloys

**DOI:** 10.3390/ma11122378

**Published:** 2018-11-26

**Authors:** Zheng-Xue Zhao, Zhen-Ming Hua, Da-Wei Li, Dong-Song Wei, Yan Liu, Jin-Guo Wang, Dan Luo, Hui-Yuan Wang

**Affiliations:** 1Key Laboratory of Automobile Materials of Ministry of Education & School of Materials Science and Engineering, Nanling Campus, Jilin University, No. 5988 Renmin Street, Changchun 130025, China; zhaozhengx16@163.com (Z.-X.Z.); huazm18@mails.jlu.edu.cn (Z.-M.H.); dwli16@mails.jlu.edu.cn (D.-W.L.); 13194360919@189.cn (D.-S.W.); jgwang@jlu.edu.cn (J.-G.W.); wanghuiyuan@jlu.edu.cn (H.-Y.W.); 2Key Laboratory of Bionic Engineering (Ministry of Education), Jilin University, Changchun 130025, China; luod15@163.com; 3International Center of Future Science, Jilin University, Changchun 130012, China

**Keywords:** magnesium alloys, microstructure, mechanical properties, corrosion behavior

## Abstract

The microstructure, mechanical properties and corrosion behavior of hot–rolled Mg–xSn–1Zn–0.5Ca (x = 1, 3 and 5 wt.%) alloys were investigated for possible application as biodegradable implants. The hot–rolled Mg–xSn–1Zn–0.5Ca alloys consisted of α-Mg matrix and Mg_2_Sn phase. The number of the Mg_2_Sn particles significantly increased and the grains were gradually refined (14.2 ± 1.5, ~10.7 ± 0.7 and ~6.6 ± 1.1 μm), while the recrystallized fraction significantly decreased with the increase in the Sn content, the Mg–1Sn–1Zn–0.5Ca alloy was almost completely recrystallized. Ultimate tensile strength (UTS) and tensile yield strength (TYS) increased slightly, reaching maximum values of 247 MPa and 116 MPa, respectively, for the Mg–5Sn–1Zn–0.5Ca alloy, and the elongation decreased with the increase in the Sn content; the Mg–1Sn–1Zn–0.5Ca alloy showed the highest elongation (15.3%). In addition, immersion tests and electrochemical measurements in Hank’s solution revealed that the corrosion rates of Mg–xSn–1Zn–0.5Ca alloys increased with the increase in the Sn content. A model of the corrosion behavior was discussed for hot–rolled Mg–xSn–1Zn–0.5Ca alloys in Hank’s solution. Among the Mg–xSn–1Zn–0.5Ca (x = 1, 3 and 5 wt.%) alloys, Mg–1Sn–1Zn–0.5Ca alloy exhibits optimal corrosion resistance and appropriate mechanical properties.

## 1. Introduction

Magnesium alloys have been regarded as a potential biodegradable metallic materials in recent years because of their similar elastic modulus and density to human bone (41–45 GPa) [1,2]. Magnesium is an essential element which participates in many biological mechanisms [3,4]; as a consequence, it is also biocompatible, bioactive [5,6]. In addition, compared to other biomedical metallic materials, for instance, stainless steel and Ti alloys, magnesium alloys are biodegradable [7,8].

Nevertheless, the inadequate corrosion resistance of magnesium and magnesium alloys has hindered their wide application as implant materials. The rapid degradation rates of Mg alloys lead to the generation of pH changes and gas bubbles, which will be harmful to the healing process [2]. More seriously, the rapid degradation rates result in a dramatic strength reduction before healing, therefore improvement of the corrosion properties of magnesium alloys is of great significance for the clinical application of biodegradable magnesium alloys.

To address the problem, many methods have been employed to improve the corrosion properties of Magnesium alloys. The first method is alloying elements into magnesium and magnesium alloys to improve the corrosion properties, for example, Mg–Mn alloy [9], Mg–Zn–Mn–xCa alloys [10], Mg–7Sn alloy [11], Mg–Zn alloy [12]. Yuan et al. [13] systematically studied a degradable Mg–Nd–Zn–Zr alloy, which shows high corrosion resistance and suitable mechanical properties because of the addition of Neodymium (Nd), Zinc (Zn)and Zirconium (Zr) elements to pure magnesium. Andrej Atrens et al. [14,15,16] believe that high purity magnesium has a higher corrosion resistance than magnesium alloys because high purity magnesium contains only α-Mg. However, high purity magnesium has relatively low mechanical properties; as a consequence, the mechanical properties and corrosion resistance of the magnesium can be simultaneously improved by adding an appropriate amount of alloying elements. The second method is forming, causing plastic deformation, such as rolling, extrusion, forging, etc. Plastic deformation can refine grain and change the distribution of second phases to improve mechanical properties and corrosion resistance simultaneously [17,18]. Recently, Cheng et al. [19] reported that the corrosion resistance of Mg–6Bi–2Sn alloy was improved by rolling because of grain refinement and the presence of passivity film. The third method is surface modification, such as plasma electrolytic oxidation coating [20], micro-arc oxidation coating [21], and zeolite coating [22], etc. Although surface modification initially exhibits high corrosion resistance, unfortunately, the surface layer is usually unstable.

Therefore, in this paper, we combined element alloying with plastic deformation (hot rolling) to improve mechanical properties and corrosion properties of magnesium alloys. For alloying element selection, there are several principles in developing biodegradable Mg alloys. The first requirement is nontoxicity, and the second requirement is that the elements have good strengthening ability [23]. Based on the above principles, Tin (Sn), Zinc (Zn) and Calcium (Ca) were chosen. Tin (Sn) is an element needed in the human body, and it has been regarded as a relatively non-toxic material [24]. In addition, it was reported that adding a moderate amount of Sn to magnesium alloys could notably improve the corrosion resistance of magnesium alloys [25]. Zinc (Zn) is also an element needed for the human body and it can refine grain and therefore improve the strength of magnesium; moreover, researches have indicated that adding appropriate Zn to pure magnesium can improve the corrosion resistance because Zn can improve the corrosion potential of Mg alloys [26,27]. It is generally accepted that Calcium (Ca) is essential element for the human body because it occurs naturally in the human metabolism, especially in the bones [28,29]. In addition, adding Ca to magnesium alloys is in favor of refining grain and improving corrosion resistance [30,31].

Most of the previous studies have been concerned with mechanical properties and corrosion properties of Mg–Sn series alloys. However, the addition of Zn and Ca elements to Mg–Sn alloys and the hot rolling deformation of the Mg–Sn series alloys are relatively rare. As a consequence, the hot–rolled Mg–xSn–1Zn–0.5Ca alloys have been developed for possible application as biodegradable implants. In this study, the effect of Sn addition on the microstructure, mechanical properties and corrosion behavior of hot–rolled Mg–xSn–1Zn–0.5Ca (x = 1, 3 and 5 wt.%) alloys has been investigated by tensile tests, electrochemical measurements and immersion tests, and a model of the corrosion behavior of as-prepared alloys was discussed to establish a correlation between corrosion behavior and microstructure.

## 2. Experimental Procedures

### 2.1. Material Preparation

Mg–xSn–1Zn–0.5Ca (x = 1, 3 and 5 wt.%) alloys were prepared using pure magnesium (99.85%), pure tin (99.90%), pure zinc (99.90%), and Mg–30Ca master alloy by an electric resistance furnace under CO_2_ + 1 vol% SF_6_ at 680 °C. To ensure homogenization, the melt was stirred 2 times for 5 min each time and kept at 30 min at 680 °C. Finally, the melt was cast into a preheated copper mould at 200 °C with a diameter of 90 mm and a height of 100 mm that contained a cavity size of 100 mm × 50 mm × 6 mm, and the castings were cooled to room temperature at atmosphere. Cast billets were solutioned at 345 °C for 4 h followed by 480 °C for 8 h in a tube furnace under a nitrogen gas, and then the samples were quenched in hot water (50 ± 5 °C). The ingots were cut into several sheets with a geometric size of 25 mm × 10 mm × 5 mm.

The sheets were rolled from 5 mm to 2 mm after nine passes. Reduction ratio of per pass was about 8%, and the rollers were preheated to 100 °C. The sheets were preheated at 360 °C for 15 min before the first pass and then were preheated at 300 °C for 10 min. Finally, the hot-rolled specimens were annealed at 275 °C for 1.5 h. The chemical compositions of each Mg–xSn–1Zn–0.5Ca sample were analyzed using an optical spectrum analyzer (ARL 4460, Ecublens, Switzerland), the results of which are shown in Table 1.

### 2.2. Microstructure Characterization

The specimens were polished and etched in acetic picral (3 mL acetic acid, 5 g picric acid, 1 mL distilled water, 20 mL ethanol) for 10–20 s. The microstructure of the hot-rolled Mg–xSn–1Zn–0.5Ca (x = 1, 3 and 5 wt.%) alloys was analyzed using an optical microscope (OM, Carl Zeiss Axio Imager A2m, Göttingen, Germany) and a scanning electron microscope (SEM, VEGA3 XMU, TESCAN, Brno, Czech) with an energy dispersive spectrometer (EDS) analyzer. The phases of specimens were measured by X-ray diffraction (XRD, D/Max 2500PC, Rigaku, Tokyo, Japan) with Cu Kα radiation at 40 KV (scanning range (2θ): 20–80°, scanning rate: 3° min^−1^, step size: 0.02°). JCPDS-International Center for Diffraction Data (JCPDS-ICDD) were applied to analyze the XRD data. The electron backscattered electron (EBSD) analysis was conducted by a scanning electron microscope with an Oxford Instruments NordlysNano EBSD detector (Instruments, Oxford, UK). The data were collected and analyzed by AZtec and Channel 5.0 software (Aztec Software Private Limited, Bengaluru, Karnataka). EBSD characterization was conducted at 20 kV, 70° tilt and a step size of 0.4 µm. The average grain size was calculated using the linear intercept method according to the ASTM E112-96 standard with at least eight SEM micrographs.

### 2.3. Tensile Tests

Tensile testing specimens were cut from the rolling direction of hot-rolled slabs. The dog bone-shaped tensile testing specimens had a dimension of 30 × 10 × 1.2 mm^3^ with a gauge size of 10 × 4 × 1.2 mm^3^. Tensile tests were conducted at room temperature using a MTS-810 tensile testing machine at an initial strain rate of 1.0 × 10^−3^ s^−1^. The elastic modulus of the specimens was analyzed by an ultrasonic pulse echo technique. The velocities of longitudinal wave and shear wave were measured using an ultrasonic thickness gauge (Olympus 38DL PLUS, Waltham, MA, USA) [32]. Each sample was measured at least five times to obtain good repeatability.

### 2.4. Electrochemical Test 

Electrochemical tests were carried out at 37 ± 0.5 °C in a container containing 250 mL Hank’s solution (Table 2) by an electrochemical workstation (GAMRY Reference 600, Warminster, PA, USA). A three-electrode cell was used for electrochemical measurements with the as-prepared specimen as the working electrode (specimens with an exposed area of 1 cm^2^), a saturated calomel electrode as the reference electrode and a platinum plate as the counter electrode. For electrochemical tests, the as-prepared samples were immersed into Hank’s solution for 1800 s to obtain an approximate steady open circuit potential (OCP). Appendix A (in the Appendix A) shows the open circuit potential of Mg–xSn–1Zn–0.5Ca alloys in Hank’s solution. The potentiodynamic polarization test was conducted at a constant scanning rate of 0.5 mV/s, starting potential −500 mV and finishing potential 500 mV relative to the OCP and then corrosion potential (E_corr_), corrosion current density (I_corr_) and cathode Tafel slopes (β_c_) were obtained by Tafel extrapolation. Appendix A (in the Appendix A) shows the specific method obtaining corrosion current density. Furthermore, the electrochemical impedance spectroscopy spectra (EIS) were determined between 100 kHz and 10 mHz. The experimental data were analyzed by fitting data with the Zsimpwin software (EChem Software, Ann Arbor, MI, USA). Each electrochemical test was duplicated to obtain very good reproducibility.

### 2.5. Immersion Test

The corrosion behavior of the hot-rolled Mg–xSn–1Zn–0.5Ca alloys was investigated by immersion tests, weight-loss experiments and hydrogen volume measurements. The samples with a size of 15 mm × 10 mm × 2 mm were ground with SiC papers up to 2000 grit, followed by the samples being immersed in a beaker containing 250 mL Hank’s solution at 37 ± 0.5 °C for 7 days. After the test, the specimens were cleaned using chromate acid (2% AgNO_3_ and 20% CrO_3_) to remove the corrosion products. The H_2_ evolution volume of the samples was measured during 7 days of immersion in Hank’s solution. According to the method proposed by Song et al. [33], we carried out the experiment of H_2_ volume measurements. Besides, corroded surfaces of hot-rolled Mg–xSn–1Zn–0.5Ca alloys were studied by SEM, EDS and XRD.

## 3. Results and Discussion 

### 3.1. Microstructure

Figure 1 shows the microstructures of the hot-rolled Mg–xSn–1Zn–0.5Ca (x = 1, 3 and 5 wt.%) alloys. The recrystallized α-Mg grain has sizes of ~14.2 ± 1.5, ~10.7 ± 0.7 and ~6.6 ± 1.1 μm on average for the Mg–xSn–1Zn–0.5Ca (x = 1, 3 and 5 wt.%) alloys, respectively. That is to say, grain size gradually decreases with the increase in the Sn content. Besides, SEM micrographs reveal that there are second phases in the matrix. Note that there is almost no second phase in the grain interiors of Mg–1Sn–1Zn–0.5Ca alloy (Figure 1a), and the number of the second phases obviously increases with increasing Sn content from 1 to 5 wt.%.

Figure 2 presents XRD patterns of hot-rolled Mg–xSn–1Zn–0.5Ca; it is observed that the Mg–xSn–1Zn–0.5Ca alloys are composed of α-Mg and Mg_2_Sn intermetallic. Note that the diffraction peaks of the Mg_2_Sn were difficult to identify for Mg–1Sn–1Zn–0.5Ca alloy, indicating that there are very few Mg_2_Sn phases in this alloy. Recent research [33] has revealed that the α-Mg and Mg_2_Sn phases have different electrode potentials, which can accelerate the corrosion of α-Mg because of micro-galvanic corrosion.

Figure 3 presents the EBSD orientation maps of the hot-rolled Mg–xSn–1Zn–0.5Ca alloys. As can be seen in Figure 3, the grain size of the hot-rolled Mg–xSn–1Zn–0.5Ca alloys decreased with increasing Sn content, which is consistent with the observation from SEM (Figure 1). The reason is that the fine Mg_2_Sn particles played a pinning effect, limiting the growth of the grain. Park et al. [34] investigated the influence of Mg_2_Sn on the microstructure evolution and dynamic recrystallization (DRX) behavior and they noted that the grains are gradually refined with increasing Sn content because the second phases (Mg_2_Sn) can act as nucleation sites for dynamic recrystallization. Moreover, it can be seen from the corresponding inverse pole figures, the hot-rolled Mg–1Sn–1Zn–0.5Ca alloy produced a much stronger basal texture.

Furthermore, to analyze the change of recrystallized fractions, the mappings of different grains types are presented in Figure 4, in which the recrystallized grains are shown in blue color, grains with substructures are shown in yellow color, and the highly deformed grains are shown in red color. The result suggests that Mg–1Sn–1Zn–0.5Ca alloy almost completely recrystallized, and recrystallized fraction in the hot-rolled Mg–xSn–1Zn–0.5Ca alloys with addition of 1, 3, 5 wt.% Sn is 99.6%, 86.5%, 46%, respectively. As it is known, second phases play a significant role in recrystallization. Huang et al. reported that fine dispersoids with diameter less than 1 µm tend to slow down grain growth and recrystallization due to the effect of Zener drag [35,36]. However, large particles with diameter exceeding 1 µm can promote recrystallization due to particle stimulated nucleation (PSN) [37]. Therefore, it is safe to say that the recrystallization fractions decreased with increasing Sn content because fine Mg_2_Sn particles inhibit recrystallization.

### 3.2. Mechanical Properties

Figure 5 shows the tensile engineering stress-strain curves of the hot-rolled Mg–xSn–1Zn–0.5Ca (x = 1, 3 and 5 wt.%) alloys. The average tensile properties are showed in Table 3. It can be seen that as the Sn content increased from 1 to 5 wt.%, the UTS and TYS slightly increased from ~223 and ~109 MPa to ~247 and ~116 MPa, respectively. The hot-rolled Mg–5Sn–1Zn–0.5Ca sheet presents the highest σ_0.2_ (116 MPa) and σ_b_ (247 MPa), mainly because the grain refinement and Mg_2_Sn particles can inhibit dislocation motion according to the Hall–Petch relationship [38] and Orowan equation [39], respectively. Note that the elongation of Mg–xSn–1Zn–0.5Ca alloys decreased with the increase in the Sn content. The Mg–5Sn–1Zn–0.5Ca alloy shows a lower elongation (~10.8%) compared to the 15.3% of the Mg–1Sn–1Zn–0.5Ca alloy. This change is primarily ascribed to premature fracture caused by the presence of coarse Mg_2_Sn particles in the hot-rolled Mg–5Sn–1Zn–0.5Ca alloy (Figure 1c). During tensile deformation, it is possible that high stress concentration produced around coarse Mg_2_Sn particles eventually leads to crack initiation and fracture. Besides, the Young’s modulus of Mg–xSn–1Zn–0.5Ca alloys increased with increasing Sn content. It is observed that the Young’s modulus of Mg–xSn–1Zn–0.5Ca alloys is similar to that of natural bone. Moreover, the Young’s modulus of Mg–1Sn–1Zn–0.5Ca alloy is closer to that of human bone.

### 3.3. Corrosion Properties in Hank’s Solution

Figure 6 shows the weight loss rate of the samples immersed in Hank’s solution at 37 ± 0.5 °C. It was observed that the weight loss rate of Mg–xSn–1Zn–0.5Ca (x = 1, 3 and 5 wt.%) alloys increased with increasing Sn content. The weight loss rate was 1.248 ± 0.072, 1.56 ± 0.06, and 3.288 ± 0.144 mg/(cm^2^ day) for the Mg–xSn–1Zn–0.5Ca (x = 1, 3 and 5 wt.%) alloys, respectively. The average corrosion rate (R_w_, mm/year) was obtained from the weight loss rate (C_R_, mg/(cm^2^ day) by the followed equation [12]:R_w_ = 2.1C_R_(1)

The corresponding corrosion rates are 2.621 ± 0.151, 3.276 ± 0.126, and 6.905 ± 0.302 mm/year for the Mg–xSn–1Zn–0.5Ca (x = 1, 3 and 5 wt.%) alloys, respectively. The hot-rolled Mg–1Sn–1Zn–0.5Ca alloy exhibits relatively optimal corrosion resistance among the Mg–xSn–1Zn–0.5Ca alloys. The main reason for this phenomenon is that the local corrosion occurred more frequently with increasing Sn content.

Figure 7 shows the variation of the H_2_ evolution volume (HEV) in Hank’s solution; results indicated that the H_2_ evolution volume increased with increasing Sn content. The results are consistent with the results of weight loss (Figure 6). In the first 70 h, the HEV gradually increases for all samples but the Mg–5Sn–1Zn–0.5Ca alloy showed greater slope value (d(HEV)/dt) than the other two alloys. From 70 h to 168 h, it is seen that HEV slows down and reaches a steady state for Mg–1Sn–1Zn–0.5Ca and Mg–3Sn–1Zn–0.5Ca alloys; the Mg–3Sn–1Zn–0.5Ca alloy still has a higher hydrogen evolution volume compared to Mg–1Sn–1Zn–0.5Ca alloy. However, in terms of Mg–5Sn–1Zn–0.5Ca alloy, the HEV still increased; this result indicated that the Mg–5Sn–1Zn–0.5Ca alloy had undergone a severe corrosion in Hank’s solution. The Mg_2_Sn phase and α-Mg have different potentials, which resulted in galvanic corrosion and then accelerated the dissolution of α-Mg. The Mg–5Sn–1Zn–0.5Ca alloy has the largest number of Mg_2_Sn phase particles among the hot-rolled Mg–xSn–1Zn–0.5Ca (x = 1, 3 and 5 wt.%) alloys. As a consequence, the Mg–5Sn–1Zn–0.5Ca alloy presents the worst corrosion resistance. It must also be mentioned that the HEV of Mg–1Sn–1Zn–0.5Ca and Mg–3Sn–1Zn–0.5Ca alloys is relatively stable after immersion 90 h; this is mainly because the corrosion products of the protective film inhibit the further corrosion of the alloys. Liu et al. [11] reported that the dissolved Sn in α-Mg matrix participated in the formation of corrosion products during the immersion test. It was observed that the corrosion products consisted of Mg(OH)_2_, SnO_2_ and compounds containing Ca/P according to the XRD(Appendix A (in the Appendix A)) and EDS. SnO_2_ will enhance the compactability of corrosion products [40].

The polarization curves of Mg–Sn–Zn–Ca alloys are presented in Figure 8a, and Table 4 shows the parameters of the E_corr_, I_corr_ and β_c_ of the samples obtained from the polarization curves by Tafel extrapolation. It can be observed that the I_corr_ increased with increasing Sn content throughout the immersion process in Figure 8a and Table 4. It can be concluded that the corrosion rates of the alloys could be ranked in an increasing series as: Mg–1Sn–1Zn–0.5Ca < Mg–3Sn–1Zn–0.5Ca < Mg–5Sn–1Zn–0.5Ca. The results were in good consistent with the weight loss and hydrogen evolution results.

To further investigate the corrosion characteristics of the alloys, Electrochemical Impedance Spectroscopy (EIS) was performed. The EIS spectra of the three alloys have a capacitive loop in the high frequency region and an inductive loop in the low frequency region. Note that the diameter of the capacitive loop in the Figure 8b increased with increasing Sn content in all frequency ranges. The same regularity was observed in Figure 8c; |Z| of hot-rolled Mg–xSn–1Zn–0.5Ca alloys increased with the decrease of Sn content, indicating that hot-rolled Mg–1Sn–1Zn–0.5Ca alloy showed the optimal corrosion properties. Besides, the equivalent circuit and the fitting data of the EIS spectra were analyzed in Appendix A (Figure 8d and Appendix A (in the Appendix A)). Historical records showed that Mg–5Sn alloy exhibits better corrosion properties than pure Mg because an Sn-rich layer is formed between passive film and matrix [25]. As a result, we can conclude that adding an appropriate amount of Sn to Mg alloys can improve their corrosion resistance.

### 3.4. Corrosion Behavior

Figure 9 shows SEM micrographs of the corroded surface of the Mg–xSn–1Zn–0.5Ca alloys after 7-day immersion in Hank’s solution. It is observed that blank and white corrosion products covered the Mg–xSn–1Zn–0.5Ca alloys surface. The surface of the Mg–5Sn–1Zn–0.5Ca alloy had more corrosion products than the other Mg–xSn–1Zn–0.5Ca alloys, indicating that the Mg–5Sn–1Zn–0.5Ca alloy exhibited a lower corrosion resistance than the other Mg–xSn–1Zn–0.5Ca alloys. To further determine the composition and phase of the corrosion products, we conducted EDS (Figure 10) and XRD (Appendix A) on the corroded surface of Mg–5Sn–1Zn–0.5Ca alloy after immersion 7 days in Hank’s solution. The results showed that the blank corrosion products mainly were Mg(OH)_2,_ and white corrosion products were compounds containing Ca/P.

Figure 11 shows the surface morphologies after removing corrosion products; it can be seen that the number of the corrosion pits increases with the increase in the Sn content. The surface of the Mg–5Sn–1Zn–0.5Ca alloy presents plenty of deep corrosion, indicating that it suffered more serious corrosion than the other Mg–xSn–1Zn–0.5Ca (x = 1 and 3 wt.%) alloys. The Mg_2_Sn phase and α-Mg matrix have different potentials; the Mg_2_Sn phase has higher corrosion potential than α-Mg matrix [32]. The Mg_2_Sn phase is used as the cathode and the α-Mg matrix as the anode, resulting in galvanic corrosion and then accelerating the dissolution of α-Mg in simulated body fluid. As a result, the increase in the Mg_2_Sn phase will aggravate the corrosion of Mg–xSn–1Zn–0.5Ca alloys, thus the local corrosion occurred more frequently with increasing Sn content.

Figure 12 shows the corrosion behavior model of the hot-rolled Mg–xSn–1Zn–0.5Ca alloys. Mg–xSn–1Zn–0.5Ca alloys consisted of α-Mg matrix and Mg_2_Sn particles. After rolling and annealing, the second phases uniformly dispersed in the α-Mg matrix. The corrosion occurs first around the second phases because of the different corrosion potential between the α-Mg matrix and Mg_2_Sn phase; the α-Mg matrix acts as an anode and Mg_2_Sn phase as a cathode during immersion. In Hank’s solution, corrosion of magnesium alloys includes cathodic reduction reactions and anodic dissolution of Mg alloys. In the first stage, the α-Mg matrix was dissolved, at the same time, an Mg(OH)_2_ layer is formed on the surface of magnesium alloys (4), (Figure 12a). In the second stage, when magnesium alloys covered by the Mg(OH)_2_ layer, the Mg(OH)_2_ layer can inhibit the further corrosion to the metal surface. Nevertheless, the Mg(OH)_2_ layer is unstable and easy to breakdown, and chloride ions are adsorbed on the Mg(OH)_2_ layer and be slightly dissolved (5), (Figure 12b). In the third stage, lots of Mg^2+^ dissolved in Hank’s solution, and the pH value gradually increased because of the cathodic reaction [41]. With the increase in the immersion time, ions of the Hank’s solution such as PO_4_^3−^ and Ca^2+^ were deposited onto the surface of the samples, resulting in Ca/P deposition on the surface (6), (Figure 12b).
Stage i: Mg (s) → Mg^2+^ (aq) + 2e^−^ (anodic reaction)(2)
2H_2_O (aq) + 2e^−^→ 2OH^−^ (aq) + H_2_ (cathodic reaction)(3)
Mg (s) + 2H_2_O (aq) = Mg(OH)_2_ (s) + H_2_ (g) (overall product formation)(4)
Stage ii: Mg(OH)_2_ (s) + 2Cl^−^ (aq) ⇄ MgCl_2_ (s) + 2OH^−^ (aq)(5)
Ca/P deposition(6)

## 4. Conclusions

Hot–rolled Mg–xSn–1Zn–0.5Ca (x = 1, 3 and 5 wt.%) alloys were developed as potential biodegradable medical metal materials, and the effect of Sn content on the microstructure, mechanical properties and corrosion behavior of hot-rolled alloys was investigated. The microstructure of hot–rolled Mg–xSn–1Zn–0.5C consisted of α-Mg matrix and Mg_2_Sn phases. The grain size was refined and the number of the Mg_2_Sn particles increased with the increase in the Sn content; the average grain size of Mg–5Sn–1Zn–0.5Ca alloy was about 6.6 μm. The recrystallized fraction significantly decreased with the increase in the Sn content, which can primarily be attributed to the Mg_2_Sn particles slowing down grain growth and recrystallization. 

Furthermore, the UTS and TYS slightly decreased; nevertheless, the elongation of the Mg–xSn–1Zn–0.5Ca alloys increased with decreasing Sn content, reaching a maximum value of 15.3% for the Mg–1Sn–1Zn–0.5Ca alloy. 

The corrosion rates of Mg–xSn–1Zn–0.5Ca alloys significantly increased with increasing Sn content; the increase of corrosion rate is mainly ascribed to the enhancement of micro-galvanic corrosion caused by the increase of the number of Mg_2_Sn particles. As a result, the number of second phases has a greater effect on corrosion properties than grain size. Corrosion of hot–rolled Mg–xSn–1Zn–0.5Ca alloys was started from local corrosion, and the amount of pits in the Mg–1Sn–1Zn–0.5Ca alloy was significantly lower than those of the Mg–3Sn–1Zn–0.5Ca and Mg–5Sn–1Zn–0.5Ca alloys. Overall, the above results show that adding an appropriate amount of Sn is a promising route towards improving the mechanical properties and corrosion resistance of Mg-based biodegradable metals.

## Figures and Tables

**Figure 1 materials-11-02378-f001:**
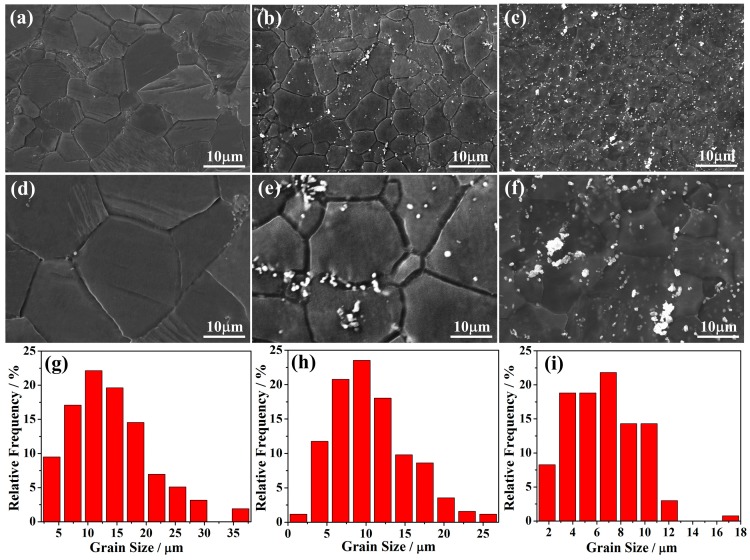
SEM micrographs of (**a**) Mg–1Sn–1Zn–0.5Ca, (**b**) Mg–3Sn–1Zn–0.5Ca, (**c**) Mg–5Sn–1Zn–0.5Ca, the corresponding magnified images (**d**–**f**), and the grain size distribution of (**g**) Mg–1Sn–1Zn–0.5Ca, (**h**) Mg–3Sn–1Zn–0.5Ca, (**i**) Mg–5Sn–1Zn–0.5Ca.

**Figure 2 materials-11-02378-f002:**
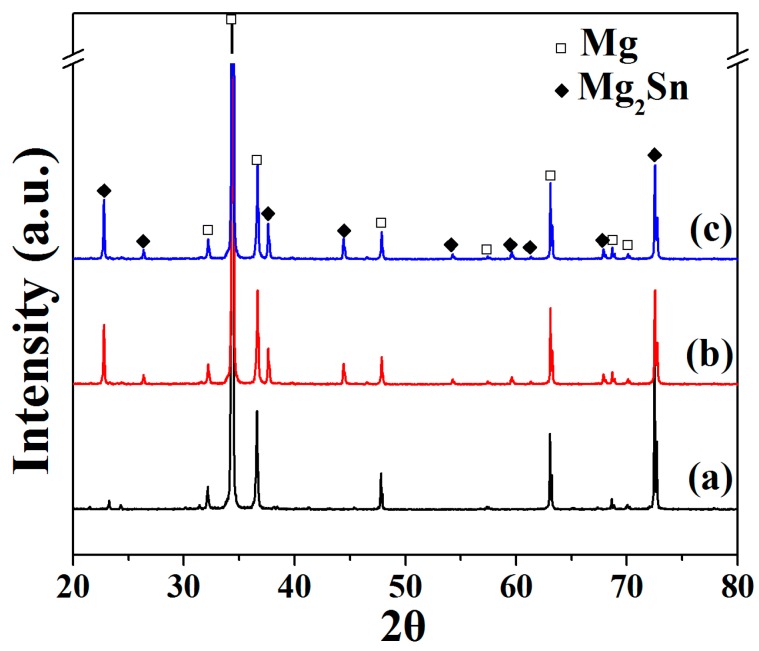
X-ray diffraction spectra of hot-rolled Mg–xSn–1Zn–0.5Ca alloys: (**a**) Mg–1Sn–1Zn–0.5Ca; (**b**) Mg–3Sn–1Zn–0.5Ca; (**c**) Mg–5Sn–1Zn–0.5Ca.

**Figure 3 materials-11-02378-f003:**
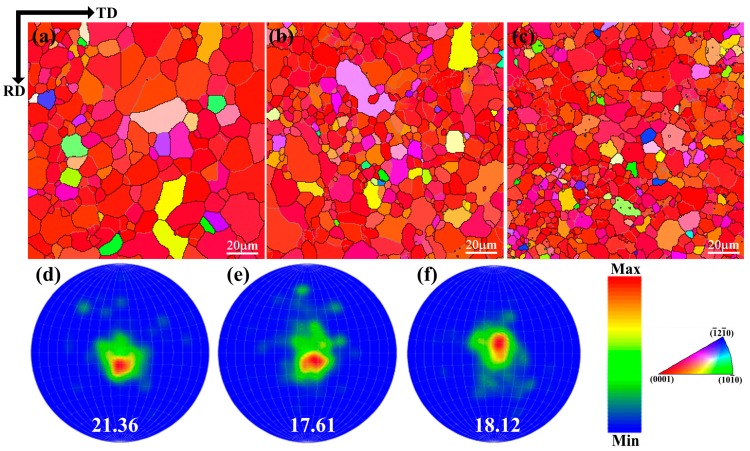
EBSD orientation maps and (0002) pole figures of (**a**,**d**) Mg–1Sn–1Zn–0.5Ca, (**b**,**e**) Mg–3Sn–1Zn–0.5Ca and (**c**,**f**) Mg–5Sn–1Zn–0.5Ca samples.

**Figure 4 materials-11-02378-f004:**
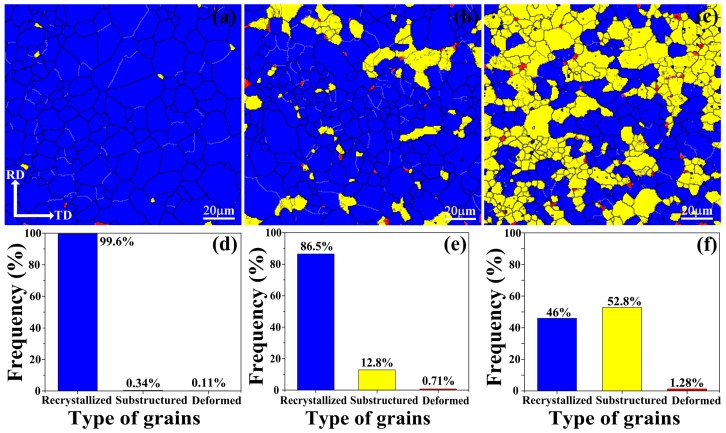
Different grains types of hot-rolled (**a**) Mg–1Sn–1Zn–0.5Ca, (**b**) Mg–3Sn–1Zn–0.5Ca, (**c**) Mg–5Sn–1Zn–0.5Ca, alloys and corresponding frequency of the different grains types (**d**–**f**).

**Figure 5 materials-11-02378-f005:**
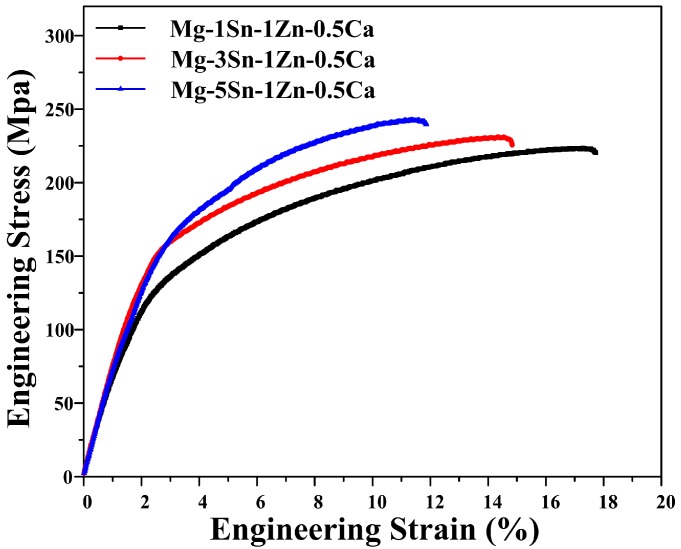
Tensile engineering stress-strain curves of hot-rolled Mg–xSn–1Zn–0.5Ca (x = 1, 3 and 5 wt.%) alloys.

**Figure 6 materials-11-02378-f006:**
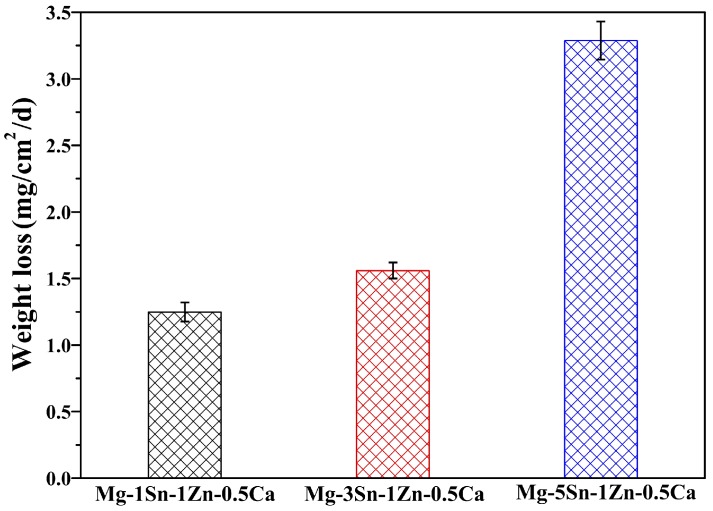
Weight loss rate of Mg–xSn–1Zn–0.5Ca (x = 1, 3 and 5 wt.%) alloys in the Hank’s solution for 7 days.

**Figure 7 materials-11-02378-f007:**
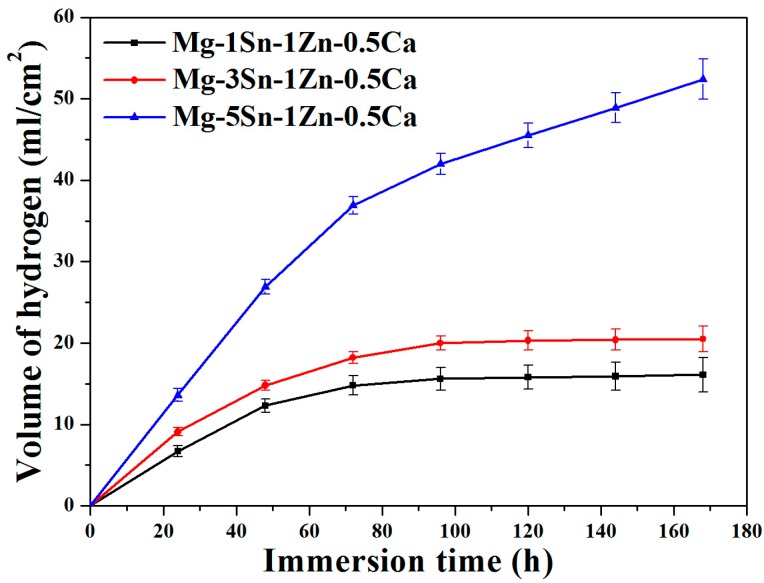
Hydrogen evolution of Mg–xSn–1Zn–0.5Ca (x = 1, 3 and 5 wt.%) alloys in the Hank’s solution for 7 days.

**Figure 8 materials-11-02378-f008:**
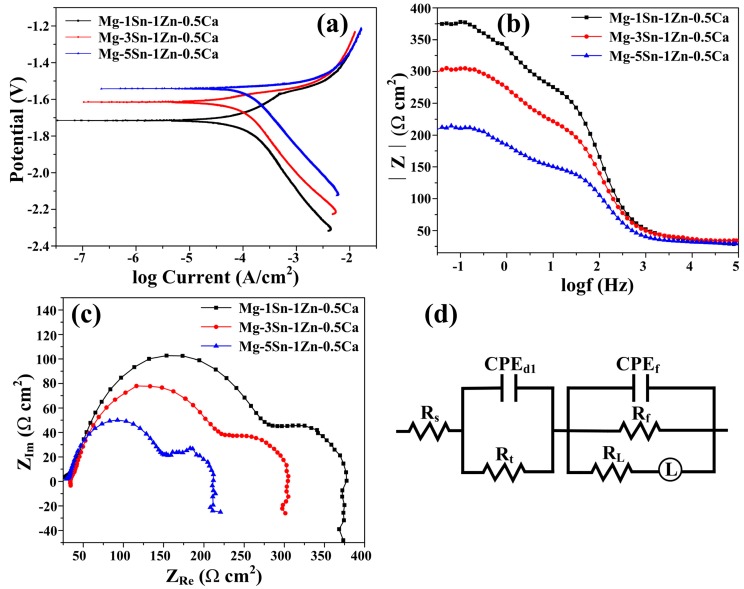
Polarization curves and Electrochemical Impedance Spectroscopy spectra of hot-rolled Mg–xSn–1Zn–0.5Ca (x = 1, 3 and 5 wt.%) alloys in Hank’s solution: (**a**) Polarization curves; (**b**) Nyquist plots; (**c**) Bode plots; (**d**) equivalent circuit.

**Figure 9 materials-11-02378-f009:**
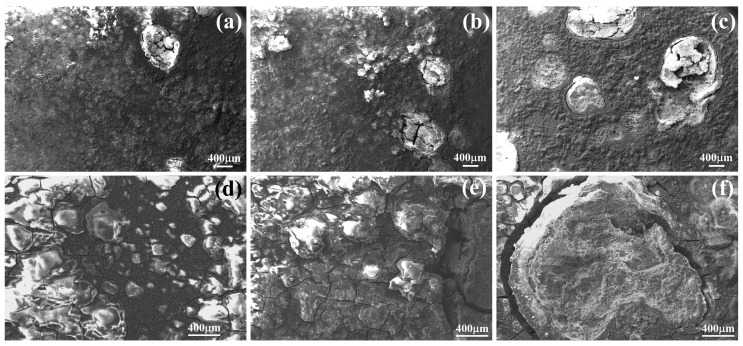
Corroded surface morphologies of hot-rolled Mg–xSn–1Zn–0.5Ca alloys after being immersed in Hank’s solution for 7 days: (**a**) Mg–1Sn–1Zn–0.5Ca; (**b**) Mg–3Sn–1Zn–0.5Ca; (**c**) Mg–5Sn–1Zn–0.5Ca, and the corresponding magnified images (**d**–**f**).

**Figure 10 materials-11-02378-f010:**
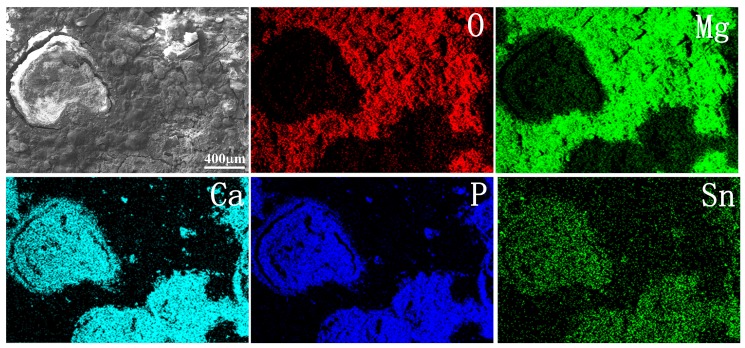
EDS spectra of hot-rolled Mg–5Sn–1Zn–0.5Ca alloy after immersion in Hank’s solution for 7 days.

**Figure 11 materials-11-02378-f011:**
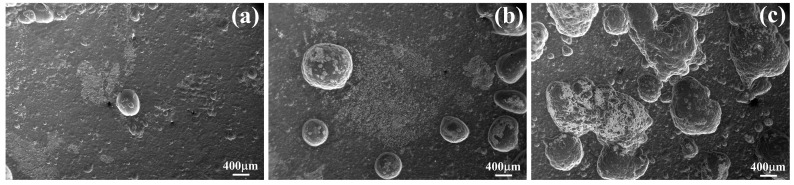
Surface morphologies after removing corrosion products: (**a**) Mg–1Sn–1Zn–0.5Ca; (**b**) Mg–3Sn–1Zn–0.5Ca; (**c**) Mg–5Sn–1Zn–0.5Ca.

**Figure 12 materials-11-02378-f012:**
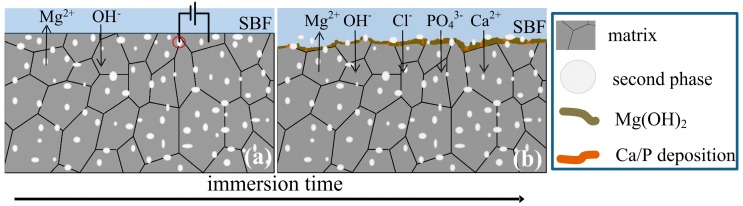
Schematic mechanism of the corrosion process of hot-rolled Mg–xSn–1Zn–0.5Ca (x = 1, 3 and 5 wt.%) alloys in simulated body fluid.

**Table 1 materials-11-02378-t001:** Chemical composition (wt.%) of the hot-rolled Mg–xSn–1Zn–0.5Ca (x = 1, 3 and 5 wt.%) alloys investigated.

Alloy	Sn	Zn	Ca	Fe	Cu	Ni	Mn	Si	Mg
Mg–1Sn–1Zn–0.5Ca	1.53	1.13	0.46	0.0015	0.0015	0.0008	0.0152	0.0109	Bal.
Mg–3Sn–1Zn–0.5Ca	2.88	1.07	0.41	0.0023	0.0020	0.0007	0.0108	0.0124	Bal.
Mg–5Sn–1Zn–0.5Ca	4.91	1.14	0.42	0.0014	0.0013	0.0007	0.0125	0.0123	Bal.

**Table 2 materials-11-02378-t002:** Compositions of the Hank’s balanced salt solution.

Reagent	Amount (g/L)
NaCl	8.00
NaHCO_3_	0.35
Na_2_HPO_4_·2H_2_O	0.48
KH_2_PO_4_	0.06
KCl	0.40
MgSO_4_·7H_2_O	0.10
CaCl_2_·2H_2_O	0.14

**Table 3 materials-11-02378-t003:** Tensile properties of the hot-rolled Mg–xSn–1Zn–0.5Ca (x = 1, 3 and 5 wt.%) alloys investigated.

Alloy	Tensile Yield Strength σ_0.2_/MPa	Ultimate Tensile Strength σ_b_/MPa	Elongation ε/%	Fracture Strain ε_b_/%	Young’s Modulus E/GPa
Mg–1Sn–1Zn–0.5Ca	$109_{-5}^{+3}$	$223_{-2}^{+4}$	$\;15.3_{-0.4}^{+0.6}$	$17.7_{-0.7}^{+2.1}$	$39.5_{-1.2}^{+0.7}$
Mg–3Sn–1Zn–0.5Ca	$113_{-6}^{+4}$	$231_{-5}^{+2}$	$12.1_{-0.7}^{+1.1}$	$14.3_{-1.1}^{+1.6}$	$40.6_{-0.4}^{+1.5}$
Mg–5Sn–1Zn–0.5Ca	$116_{-4}^{+7}$	$247_{-3}^{+1}$	$10.8_{-0.3}^{+1.2}$	$11.8_{-1.3}^{+1.7}$	$43.8_{-1.3}^{+1.1}$

**Table 4 materials-11-02378-t004:** Fitted results from polarization curves.

Alloy	E_corr_ (V)	I_corr_ (µA)	β_c_ (mV/decade)
Mg–1Sn–1Zn–0.5Ca	−1.723 ± 0.01	38.65 ± 7.28	−185.6 ± 4.9
Mg–3Sn–1Zn–0.5Ca	−1.620 ± 0.00	63.98 ± 4.15	−245.3 ± 2.6
Mg–5Sn–1Zn–0.5Ca	−1.571 ± 0.01	92.65 ± 5.83	−282.5 ± 4.3

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
