# Peer review of "Effect of Sn Content on the Microstructure, Mechanical Properties and Corrosion Behavior of Biodegradable Mg–x (1, 3 and 5 wt.%) Sn–1Zn–0.5Ca Alloys"

_materials, 2018, doi:10.3390/ma11122378_

Round 1
Reviewer 1 Report
The paper is focused on the design of a Mg-xSn-1Zn-0.5Ca biodegradable alloy, studying the influence of Sn content (1, 3 and 5% wt.) on its microstructure, mechanical and corrosion behaviour. SEM, EBSD, XRD, tensile and electrochemical testing were principally employed for the evaluation of the respective material properties. The paper is structured and well written and it could be accepted for publication, considering some additional issues and comments:
1. In Table 3, reporting the mechanical (tensile) properties, the column depicting the Elongation is vacant, while the corresponding values are reported in the text. Please check and revise.
2. Reference to Figures should be in the same order of appearance, e.g. Figure 4 is cited in L158, after Figure 1 and before Figure 2. Please check and revise.
3. In Table 4, where corrosion parameters are listed, corrosion potential seems less negative (more cathodic) for the highest Sn content alloy and more negative (more anodic) for the lowest Sn content alloy, signifying higher corrosion susceptibility for the 1% Sn alloy (the same order is depicted by Figure S1 showing the variation of OCP). Corrosion current results showed the opposite trend; please comment.
4. Figures 9 and 11 showing SEM images of corroded surfaces seem very small, as embedded in the manuscript. Especially Figure 9, which seems that smaller insets of higher magnification are inserted inside each micrograph, is barely understood and, thus, it needs enlargement.
5. Please separate, by one single space, the units from the numerical part of a physical entity (especially for mechanical properties); e.g. L203,… 116MPa to 116 MPa, and so on...
6. The language and spelling is sufficient. However, careful proofreading is necessary to eliminate minor spelling errors and edits. For instance:
- L238… have different instead of has different
- L240…. largest number of… phase particles (add the word “particles”)
- L310… magnesium alloys(3) : explain what does the number mean in the parentheses
Author Response
We have uploaded response to the reviewer’s comments as a PDF file.

Reviewer 2 Report
see additional comments

Author Response

(The authors gave the same response as above.)
